# Accessing Structural, Electronic, Transport and Mesoscale Properties of Li-GICs via a Complete DFTB Model with Machine-Learned Repulsion Potential

**DOI:** 10.3390/ma14216633

**Published:** 2021-11-03

**Authors:** Simon Anniés, Chiara Panosetti, Maria Voronenko, Dario Mauth, Christiane Rahe, Christoph Scheurer

**Affiliations:** 1Department of Chemistry, Technische Universität München, Lichtenbergstr. 4, 85748 Garching, Germany; simon.annies@tum.de (S.A.); maria.voronenko@tum.de (M.V.); dario.mauth@tum.de (D.M.); 2Fritz Haber Institute of the Max Planck Society, Faradayweg 4-6, 14195 Berlin, Germany; scheurer@fhi.mpg.de; 3ISEA, RWTH Aachen University, Jägerstraße 17-19, 52066 Aachen, Germany; christiane.rahe@isea.rwth-aachen.de

**Keywords:** lithium-ion batteries, DFTB, Li-GIC, graphite, intercalation, multiscale modeling, diffusion barriers, formation energies, energy materials, machine learning

## Abstract

Lithium-graphite intercalation compounds (Li-GICs) are the most popular anode material for modern lithium-ion batteries and have been subject to numerous studies—both experimental and theoretical. However, the system is still far from being consistently understood in detail across the full range of state of charge (SOC). The performance of approaches based on density functional theory (DFT) varies greatly depending on the choice of functional, and their computational cost is far too high for the large supercells necessary to study dilute and non-equilibrium configurations which are of paramount importance for understanding a complete charging cycle. On the other hand, cheap machine learning methods have made some progress in predicting, e.g., formation energetics, but fail to provide the full picture, including electrostatics and migration barriers. Following up on our previous work, we deliver on the promise of providing a complete and affordable simulation framework for Li-GICs. It is based on density functional tight binding (DFTB), which is fitted to dispersion-corrected DFT data using Gaussian process regression (GPR). In this work, we added the previously neglected lithium–lithium repulsion potential and extend the training set to include superdense Li-GICs (LiC_6−*x*_; x>0) and lithium metal, allowing for the investigation of dendrite formation, next-generation modified GIC anodes, and non-equilibrium states during fast charging processes in the future. For an extended range of structural and energetic properties—layer spacing, bond lengths, formation energies and migration barriers—our method compares favorably with experimental results and with state-of-the-art dispersion-corrected DFT at a fraction of the computational cost. We make use of this by investigating some larger-scale system properties—long range Li–Li interactions, dielectric constants and domain-formation—proving our method’s capability to bring to light new insights into the Li-GIC system and bridge the gap between DFT and meso-scale methods such as cluster expansions and kinetic Monte Carlo simulations.

## 1. Introduction

Lithium-graphite intercalation compounds (Li-GICs) are the primary anode material for commercial Li-ion batteries with a market share of 98% [1] due to their good volumetric and gravimetric capacities, long cycle life, abundant availability, and low cost. Despite investigations into alternatives such as lithium-metal anodes, graphite and modified-graphite compounds will not be replaced in the foreseeable future, as important EV manufacturers, material suppliers and cell producers have recently announced that graphite-containing composites will mark the state of the art for next-generation lithium-ion batteries [2].

Lithium can intercalate into graphite (in an energetically favorable way) up to a stoichiometry of LiC_6_ which is commonly taken as the compound defining the state of charge (SOC) of 100%. Recent studies [1,3] (and references therein) have shown that so-called superdense configurations (LiC_x_;x=2−6) must also be expected, at least locally, as well under ambient conditions. It has been suggested that doping may have the potential to stabilize these compounds and make them accessible for use in batteries [4].

Between SOC 0% (i.e., graphite) and 100% (LiC_6_), the system goes through multiple phase transitions [5,6] between so-called stages (n=1,2,…) that can be experimentally discriminated. Traditionally, these stages have been interpreted in an idealized structural model to directly correlate with the number of (n−1) empty layers between each pair of filled layers. According to the Daumas–Heróld domain model [7], these configurations will rather form local islands or domains of unknown size.

During the process of filling the system, the lattice parameter in the *z* direction changes from 3.355 Å per layer at SOC 0% to 3.687–3.706 Å at SOC 100% [8,9]. Additionally, at some point between 5% and 15% SOC, the graphite structure shifts from AB-stacking to AA-stacking, possibly with intermediate configurations such as AAB or ABC [10,11].

Several characteristics make this system challenging to simulate: firstly, layers filled with Li-ions are governed by electrostatics, whereas empty layers are governed by van der Waals (vdW) interactions. Any reliable model must be able to treat both accurately. Secondly, properties such as domain sizes and low-SOC phenomena require large supercells to be investigated. Furthermore, thirdly, Li-GICs—in the context of Li-ion batteries—are an *active* material. Therefore, not only are energetics important, but so are transport properties such as diffusion barriers.

To date, neither a full DFT approach nor a pure machine learning (ML) approach have proven to be capable of efficiently meeting *all* of these requirements: DFT methods [12,13,14] are too computationally expensive to treat the size of supercells necessary and allow for extensive sampling, whereas pure machine learning approaches [10,15] usually only predict some of the required properties, but not all of them, since they do not grant access to electronic properties such as band structures and charge transfer. Density functional tight binding (DFTB), however, can be 2–3 orders of magnitude faster than DFT (which is comparable to, e.g., charge-adaptive force fields) while still retaining a physical description of the system’s electronic properties [16].

In this work, we thus employed a DFTB approach to calculate the structural and energetic properties necessary for a full description of the Li-GIC at all states of charge including superdense ones beyond LiC_6_. This comprises bond lengths, layer spacing, formation energetics, long-range Li–Li interactions, and diffusion barriers, all over a wide range of SOC. Our predictions compare favorably with experimental results [5,6,9,17,18,19,20,21,22,23] and state-of-the-art dispersion-corrected DFT [11,12,13,14,24,25], wherever available.

## 2. Materials and Methods

### 2.1. Computational

For this study, we used the implementation in DFTB+ [16] with the parametrization developed in our group. The corresponding Slater-Koster files are publicly available (see Data Availability Statement). The electronic parameters (in [26], only the confinement potential) were optimized by means of particle swarm optimization (PSO) [27]. In our GPrep approach, the repulsion potential was then fitted using Gaussian process regression (GPR) [28] as described in [26]. The initial parametrization in [26] did not include the Li–Li repulsion. Extending that earlier work, the training set now includes not only a wide variety of Li-GIC configurations between SOC 0% and 100%, but also molecular dynamics (MD) snapshots of lithium metal clusters (cf. Appendix A), as well as rattled structures extracted from geometry relaxation pathways of LiC_2_ and LiC_1.75_, so that our model can also be used to investigate superdense compounds as well as regions governed by metallic interactions, such as dendritic, mossy, or plated lithium. The GPrep hyperparameters were manually adjusted to reproduce selected properties. Additional details on how the potential shapes changed between [26] and the present work are provided in the Appendix A.

DFT calculations, serving as a reference for the DFTB fit, were performed with the all-electron framework FHI-aims [29] with light settings and default tier-2 basis sets, using the PBE exchange-correlation functional [30]. For dispersion correction, the MBD approach was chosen [31,32]. MD simulations for generating the training set and validation structures were performed in the NVE ensemble at 300 K and 1000 K using the LAMMPS code [33] with the embedded atom method (EAM) potential for alkali metals developed by Nichol and Ackland [34].

Geometries were constructed and analyzed by means of the atomic simulation environment (ASE [35]) which we also used as a base framework for all force and energy calculations, structure relaxations (specifically using the BFGS algorithm as an optimizer [36]), and barrier calculations. For the latter, we employed the nudged elastic band (NEB) [37,38] algorithm with the FIRE-optimizer [39] and climbing image switched on.

For all DFTB calculations, we used a well-converged k-point density of at least 0.1/Å. The SCC-tolerance is $10^{-7}$. We employed Fermi filling with a Fermi temperature of 300 K, as well as a Broyden mixer [40] for convergence acceleration with a mixing parameter of 0.5. All of these settings were tested with regard to convergence for the whole range of SOC. As described in [26], our parametrization is meant to be used with the Leonnard–Jones dispersion correction [41] switched on.

### 2.2. Experimental

Open circuit voltage (OCV) curves were recorded to compare the simulation with real measured values. For this purpose, cells were built with graphite against lithium as well as highly oriented pyrolytic graphite (HOPG) against lithium. The cells from EL-Cell (ECC-Std), comparable to button cells, were used as the housing. A Whatman GF\D was used as the separator and EC:DMC 1:1 with 1 mol/L LiPF6 from Sigma-Aldrich was used as the electrolyte. The graphite electrodes were coated on copper current collectors, whereas the HOPG was used without a current collector. The electron conductivity was sufficient due to its low current rate. The graphite was used in 18 mm blanks, whereas the HOPG was cut into narrow strips with a width of approximately 2 mm to ensure the highest possible surface-to-volume ratio. The ions can only intercalate into the HOPG from the cut edges and not through the surface. The graphite cells were initialized with a current rate of C/10 and the HOPG cells were cyclized with a current rate of C/30.

## 3. Results and Discussion

### 3.1. Structural Properties

Graphite consists of graphene sheets, within which the C-atoms are arranged in a hexagonal honeycomb structure. The sheets are stacked in an AB-stacking order. The lattice is hexagonal with a 2-layer, 4-atom unit cell and lattice parameters *a* and *c* [42]. In our first benchmark, we compared the performance of our DFTB parametrization with Gaussian process regression-based repulsion potential (GPrep-DFTB) with experimental and recent theoretical findings (Table 1).

**Table 1 materials-14-06633-t001:** Lattice parameters *a* and *c* of the AB-graphite unit cell, predicted using GPrep-DFTB, compared with experimental results, state-of-the-art dispersion-corrected DFT calculations, and a recently published machine learning model.

	GPrep-DFTB	Experiment		DFT				ML
**Method:**		**(** * **a** * **)**	**(** * **b** * **)**	**(** * **c** * **)**	**(** * **d** * **)**	**(** * **e** * **)**	**(** * **f** * **)**	**(** * **g** * **)**
a[Å]	2.476	2.464	2.461	2.468	2.465	2.477	2.472	2.461
c[Å]	6.746	6.711	6.709	6.712	6.645	7.087	6.975	7.538

Experimental references: (a) [17]; (b) [18]; DFT: (c) revPBE-D3-BJ [14]; (d) optB88-vdW [12]; (e) revPBE-vdW [12]; (f) vdW-optPBE [43]. Machine learning (ML) reference: (g) Atomistic Neural Network [10].

All considered benchmark methods performed well in reproducing the in-plane lattice parameter *a*, which is governed by covalent C–C bonds. However, even state-of-the-art dispersion-corrected DFT functionals (except for [14]) struggle with predicting the out-of-plane lattice parameter *c*. This is due to the fact that the latter is governed by van der Waals interactions, which are still notoriously difficult to account for despite considerable effort in creating various correction schemes [31,44,45,46] for DFT. The recent machine learning approach [10] overestimates *c* by an even larger margin.

GPrep-DFTB results are very close to the experimental references—closer than even the majority of DFT approaches—proving that the method is very capable of treating both covalent C–C bonds and van der Waals interactions in graphite.

When lithium intercalates into graphite, the graphene sheets shift from AB-stacking to AA-stacking somewhere between 5% and 15% SOC, and the interlayer distance expands from ~3.36 Å to ~3.62–3.7 Å (depending on the SOC of adjacent galleries) for the full gallery. Empty galleries adjacent to filled galleries also slightly expand, due to the extra charge transferred to the graphene sheet from the intercalated Li-ions, making the overall increase in the average *z* direction lattice parameter non-linear.

It is generally accepted that Li-ions do not evenly distribute throughout the entire GIC, but tend to arrange themselves in fully filled domains and empty domains [7,47] (see Figure 1), leading to a local staging behavior with the staging number (n=1,2,…) indicating that n−1 galleries are empty between each pair of filled galleries only within that limited region. For our second benchmark, we calculated the average interlayer distances depending on the stage n=1,…,9 of stoichiometry LiC_6n_ (corresponding to LiC_6_, LiC_12_, LiC_18_, LiC_24_ and higher) with the GPrep-DFTB and compared with experimental and DFT references, where available (Figure 2 and Table 2).

In order to be able to directly compare with DFT, this set of unit cells was constructed with global staging (Figure 1, left) and not according to the domain model, which would render them far too big for DFT. Because of that, it is not obvious whether AA- or AB-stacking should be assumed for the empty layers. In *real* samples, which are large of scale and governed by the domain model (Figure 1, right), it is probable that empty parts of galleries also exhibit AA-stacking, because they are forced into that configuration by adjacent filled domains within that same gallery and because they are not truly empty, either. However, in an idealized system, without factoring this in (Figure 1, left), AB-stacking of the empty galleries (as in pure graphite) is also conceivable. Therefore (and because it is unclear which stacking order has been assumed in the DFT reference [13]), we provide predictions for both as well as a prediction area (light blue). Filled galleries are always in AA-stacking.

For the empty galleries, we predict that interlayer distances remain mostly constant throughout all stages with a slight increase in stage 2, due to the additional electrostatic repulsion caused by the charge transfer from Li-ions in the adjacent filled galleries to the carbon sheets. For the filled galleries, a constant interlayer distance can be observed for stages 4 and higher, whereas stages 3, 2, and 1 progressively show increased interlayer distances, which we attribute again to the electrostatic repulsion of the increasing charge density. Based on these findings, a simple building block model that assumes invariant interlayer distances proves sufficient to describe the system’s behavior in the high-stage (n>3) limit, where filled galleries are too far apart to interact in any way (Figure 2, left: ‘fit AA’ and ‘fit AB’). Only for stages 1 and 2 can the increased filled interlayer distance cause a significant deviation from this model.

Experimental results [5,9] agree very well with our GPrep-DFTB predictions for stages 1, 2, and 3. The DFT-based Ising model by [13] is also accurate for stages 1 and 2, but maintains an overly steep slope for higher stages 3 and 4, which—if continued—would clearly converge towards wrong asymptotic behavior.

### 3.2. Diffusion Barriers

Diffusion barriers for ion transport are among the most interesting properties of mixed ion-electron conductor (MIEC) materials such as Li-GICs. They are the crucial input parameter of any kinetic Monte Carlo simulation [48] that aims to predict large-scale phenomena such as phase transitions between stages or non-equilibrium configurations during fast charging. They are also quite difficult to reliably calculate, since they are closely linked to the interlayer distance, which, as previously pointed out, is still hard to predict, even with state-of-the-art DFT, especially for dilute, low SOC configurations.

In order to rigorously investigate the transport properties of Li-GICs, we constructed a variety of structures based on 2- and 3-layered supercells with 36 and 48 carbon atoms, respectively. This allows us to consider different staging and Li-stacking orders for equivalent stoichiometries. We fully relaxed all structures, extracted the interlayer distances, and calculated the energy barriers (Table 3) for exemplary bridge-path diffusion processes (connecting the next-neighbor Li positions) using the NEB method (see Appendix A). for the exact initial and final states of the bridge path NEBs, as well as the predicted barrier diagrams.

Analyzing the interlayer distances first, there are multiple trends to observe. First of all, structures with an AαAα order (Greek letters indicating the Li layer order) consistently relax to a smaller interlayer distance than structures with an AαAβ stacking, implying that the former may be more favorable. In terms of the total energy per unit of 6 carbon atoms, however, we see virtually no difference (AαAβ: −297.279 eV/6C, AαAα: −297.253 eV/6C) for both structures fully relaxed in terms of cell parameters and all atom positions. This deviation of E(AαAα)−E(AαAβ)= 26 meV/6C is on the order of $k_{B}T$ at room temperature, implying that at ambient conditions, no clear distinction between the Li orderings can be made and experiments would probably see a mixture of both. For reference, DFT (PBE + D3), which we consider trustworthy at least for high SOC compounds, predicts a deviation of E(AαAα)−E(AαAβ)=−14 meV/6C. It is necessary to recognize the difference in sign here, but since both values are within $k_{B}T$ at room temperature and at the limit of DFT errors, we do not believe that this constitutes a relevant difference.

Furthermore, for stoichiometries which can either be arranged as dilute stage 1 or dilute stage 2 (Li_2_C_36_) geometries, stage 2 has the lower interlayer distance and is therefore favored, which is in line with the agreed upon theory of staging and domain formation [47]. This is due to the fact that the *z* direction expansion of a gallery does not vary linearly with the filling factor. The total expansion from the 0% filling to 100% filling is 0.315 Å for AαAα-stacked stage 1 compounds. Filling empty layers by 33% (stage 1—Li_2_C_36_) already expands the interlayer distance by 0.165 Å, which is 52% of the total expansion. At 66% filling (stage 1—Li_4_C_36_), the interlayer distance is expanded by 0.252 Å, which is 80% of the total expansion.

In terms of the diffusion barriers, experimental sources vary quite a lot. This is due to the fact that the measuring technique, as well as additional factors such as temperature, play a role. Furthermore, if total diffusivities are measured, it is difficult to separate those into energetic and kinetic contributions. Langer et al. [21] measured a barrier of 0.55 eV (LiC_6_) by means of Lithium nuclear magnetic resonance (Li-NMR), Freilander et al. [22] report 0.6 eV (LiC_6_) and 1.0 eV (LiC_12_) using beta-NMR and Magerl et al. [23] find 1.0 eV (LiC_6_) by means of the quasielectric neutron scattering (QENS) at T>630 K.

On the theoretical (DFT) side of things, the revPBE-D3-BJ approach by Thinius et al. [14] (which has proven very accurate for structural parameters and formation energies) predicts the barriers of 0.42 eV (LiC_6_) and 0.47 eV (LiC_12_). Toyoura et al. [24] reported 0.3 eV (LiC_6_) and 0.49 eV (empty gallery) by means of DFT (LDA) and Persson et al. [25] predict 0.283 eV (LiC_6_) and 0.297 eV (LiC_12_), using DFT (GGA) with the interlayer distances fixed at experimental values. Even though there is some variation in the absolute numbers, both theoretical and experimental studies are in general agreement on the ordering of the barrier heights: LiC_6_ < LiC_12_ < empty gallery. This corresponds to an inverse dependency on the interlayer distance of the respective gallery.

Using GPrep-DFTB, we predict barriers of 0.404 eV (LiC_6_), 0.426 eV (LiC_12_) and 0.504 eV (empty gallery). These results capture the same previously explained qualitative trend as the references. This also holds true for other configurations that had not been investigated before, such as AαAβ-stacked and stage 3 structures. Quantitatively, our results are in particularly good agreement with the revPBE-D3-BJ approach [14].

### 3.3. Formation Energetics

The intercalation energies of Li-ions entering the GIC at different states of charge are a crucial measure for predicting the most stable configurations throughout the charging process and the phase transitions between those. Therefore, as a third benchmark, we calculated the formation energies of LiC_6_, LiC_12_ and LiC_18_, and compared them with various experimental and theoretical studies (Table 4).

We calculate the intercalation energies per lithium atom (or, equivalently, per formula unit), as
(1)$$\Delta E_{\mathrm{int}}\left(\mathrm{Li}C_{6n}\right)=E\left(\mathrm{Li}C_{6n}\right)-nE\left(C_{6}\right)-E\left(\mathrm{Li}\right)$$
where *n* is the stage number and E(·) is the DFTB total energy. These are directly comparable to the corresponding literature values obtained by DFT calculations. The latter thus do not include any finite temperature effects. On the other hand, experimental values are formation-free energies or enthalpies. Additionally, the calorimetric reference [20] is taken at elevated temperature (455 K) and with liquid lithium as precursor, rather than at room temperature with a solid lithium electrode. As a final note, the calculated values correspond to infinite phases of perfect stage *n* stoichiometry, whereas the true compounds at the corresponding stoichiometries are a mixture of domains of yet unknown structural details. Consequently, the values given within the scope of this parametrization study are not yet intended to be quantitatively comparable to the experiment, but one may still identify qualitative trends, just like with regular DFT. The DFTB model opens up the way to forthcoming more realistic, quantitative simulations.

For the formation energies, the experimental studies [6,19,20] do not agree as closely, as for the structural parameters, but they do at least provide the same ordering for LiC_6_ and LiC_12_ and LiC_18_. Interestingly, the values extracted from open circuit voltage (OCV) measurements only agree in the ordering if the energies are taken per formula unit. Normalizing the energies per graphite unit, LiC_12_ would have a less negative formation energy than LiC_6_. As the OCV curves in [6,19] agree with each other, we attribute the mismatch to different methods for extracting the formation energies from the OCV curve. We also note here that our structural models closely correspond to highly oriented pyrolytic graphite (HOPG), while all the referenced OCV curves were taken with different forms of graphitic carbon. In order to make sure that the deviation was negligible, we measured the OCV curve for HOPG. Due to the small insertion surface for Li-ions, the characteristic voltage plateaus are not as visible. However, in the regions corresponding to the phase transitions between LiC_18_ to LiC_12_ and LiC_12_ to LiC_6_, the measured HOPG curve matches within 0.01 V with the references. Our measured OCV curve is shown in the Appendix A for both HOPG and the coin cell.

Similarly, as for the C–C interlayer distance, the different DFT functionals vary significantly in their performance predicting the formation energetics of Lithium-GICs. Compared with the experimental studies, revPBE-D3-BJ [14] proves to be best, just as it did previously for the C–C interlayer distance. According to [12], revPBE-vdW correctly predicts the phase transitions between stage 1 and stage 2 qualitatively (even though it strongly underestimates the formation energies), whereas optB88-vdW does not. For revPBE-D3-BJ, we do not have this information.

Overall, the formation energy for LiC_6_ tends to be consistent across experimental measurements and most computed references. Our results with GPrep-DFTB are equally accurate for LiC_6_, while for both LiC_12_ and LiC_18_, we obtain more negative formation energies than the majority of references (with the exclusion of [6]). However, this is not necessarily a pitfall, considering that the finite temperature contributions are not included. The effect of the latter is generally nontrivial; in particular, the entropy variation in Li-GICs is *negative* for the largest part of the state of charge range. Given the overall uncertainty in both experimental and computed references, we leave this question open for further investigation and adjustments to the parametrization, if needed. We note in passing that potential refinements to the GPrep-DFTB parametrization are possible with little effort, by simply retraining the repulsion potential with additional training data and/or finely tuned hyperparameters. As a perspective, we intend to use this parametrization to train a cluster expansion similarly to [10,49], in order to perform free energy sampling and calculate the OCV curve. If that agrees with the measurements, then the non-perfect energetics of single ideal geometries is only a minor setback.

### 3.4. Long-Range Interactions

Having successfully benchmarked our GPrep-DFTB approach against a variety of comparatively small-scale properties, we proceeded towards calculating some larger-scale properties which are out of reach of DFT (at least at a reasonable computational cost). First, we want to investigate the long-range in-plane interaction between two Li-ions within Li-GIC. In order to do that, we constructed a supercell with 218 carbon atoms and two layers to eliminate any periodic next-layer interactions and allow for the bulging of the graphene sheets. We then exhaustively performed 47 full structure relaxations of all symmetry-inequivalent local minima and maxima for two Li atoms within a single layer in that supercell, as well as 41 five-image NEBs for the diffusion processes between each adjacent pair of local minima. With our method, all of this is possible within days and on a regular workstation. This leaves us with 170 data-points, which we use to fit a 2D potential energy landscape for the whole supercell (Figure 3a) and also to plot the Li–Li interactions as a function of the Li–Li distances (Figure 3b,c).

As our results clearly show, the in-plane Li–Li interactions are governed by Coulombic repulsion. Even quantitatively, our predictions agree very well with the approach of Pande et al. [13] (BEEF-vdW-DFT + Ising model). However, they were only able to provide four data points which are local minima and therefore quite cheap to calculate, whereas we can also predict transition states (which require NEB calculations and are much more computationally expensive because of that). This proves that GPrep-DFTB is capable of very accurately capturing the Li-GIC system’s electrostatic properties, and of doing so for vastly more and larger supercells than DFT.

Based on this, it is then possible to extract the slope from Figure 3c, which, via the relation:(2)$$E=\frac{e^{2}Z^{2}}{4\pi \varepsilon_{r}}\;\frac{1}{R}$$
gives us access to the relative dielectric constant $\varepsilon_{r}$ of the system. We note, however, that this is the effective dielectric response experienced by Li-ions within the Li-GIC and not the macroscopic dielectric constant of the GIC *including* the contribution due to the Li ion motion. By means of linear regression, we obtain a slope of 0.996 ± 0.015eVÅ. This leads to $\varepsilon_{r}/Z^{2}$ = 14.46 ± 0.22. For an assumed partial charge *Z* of 0.8 to 0.9 for the Li-ions, which is in line with the charge analysis performed by Rana et al. [50], we then predict a relative dielectric constant of 9.1–11.9. Expanding on this in future work, we will be able to, for the first time, determine the dielectric constant of Li-GIC as a function of the state of charge, which is an important input parameter for kinetic Monte Carlo simulations of charge carriers.

### 3.5. Domains vs. Dilute

While the general truth of the Daumas–Heróld domain model [7] has been widely accepted and supported by both theoretical [51,52] and experimental [53,54] studies, qualitative details such as domain sizes and shapes, dependencies on the charging speed and other dynamic factors have not been understood to a sufficient degree. As pointed out in [55], the formation of domains is at least partially responsible for the wide range of Li-ion diffusivities reported from the experiment ($10^{-6}-10^{-14}\frac{cm^{2}}{s}$ [25]), and therefore of crucial importance for understanding the overall behavior of Li-GICs, especially when exposed to the rapidly growing charging speeds that are necessary today.

In order to provide a further demonstration of our method’s potential to bring forward a new understanding of these phenomena in the future, we constructed four supercells with roughly 600 atoms each—two of them in a dilutely spaced configuration (with the Li-ions spread evenly across the filled layers, right column in Figure 4) and two of them with a configuration according to the Daumas–Heróld domain model (Figure 4, left column)—and performed full structure relaxations on each of them. Note that what we show here is *one* possible realization of the dilute LiC_12_ and LiC_24_, but there are necessarily many other disordered realizations with very similar total energies and any experiment would likely see a mix of these.

Our results show that both in the stage 1 (upper row) and the stage 2 (bottom row) compound, the domain–structure expanded to a significantly smaller overall interlayer distance, meaning it is favored compared to the evenly spaced one. This agrees with the Daumas–Heróld domain model [7]. Furthermore, we were also capable of extracting *local* interlayer distances for different areas of the structure. As shown in Figure 4, the difference in interlayer distance between the empty and filled domains is larger than 0.2 Å, which is much larger than any residual differences in the computed values above. According to both DFT [56] and our own results (Table 3), this difference corresponds to a difference in diffusion barriers of approximately 0.1 eV or 25%. Given this direct dependency, GPrep-DFTB could, for example, be used to reliably predict local diffusivities in large structures without the need to perform costly NEB calculations. Additionally, we believe that the capacity of GPrep-DFTB for fully relaxing large unit-cells with a multitude of Li-ion distributions has great potential for building more diverse and well-rounded training-sets for, e.g., lattice gas expansions or machine-learning models, than DFT could, also including the ’empty’ or ’almost empty’ regions of the Daumas–Heróld domain model, for which we provided an extensive model in Figure 3. This makes GPrep-DFTB a crucial new bridge between the atomistic scale and the macroscopic scale of Li-GIC modeling.

## 4. Conclusions

The structural, energetic and electronic properties of Li-GICs were theoretically investigated with our DFTB parametrization (based on GPR repulsion fitting) and benchmarked against dispersion-corrected DFT calculations and experiments. The calculated lattice parameters of graphite (a=2.476 Å, c=6.746 Å) agree better with experiments than most DFT approaches. For stages 1 through 4, our method correctly predicts the non-linear nature of the increase in the interlayer distance in LiC_x_ upon intercalation, not only qualitatively but quantitively as well. LiC_6_ relaxes to an interlayer distance of 3.682 Å. The calculated formation energies of −0.14 eV (LiC_6_), −0.40 eV (LiC_12_) and −0.42 eV (LiC_18_) per formula unit slightly overestimate the experimental results, but are within the range of DFT predictions. We expect future calculations which include entropy effects to be even more accurate. The calculated diffusion barriers (0.396 eV–0.504 eV, depending on the configuration) show trends supported by accepted theory, such as the Daumas–Heróld domain model, and agree with state-of-the-art DFT studies and experiments. In terms of long-range Li–Li interactions, our model captures the Coulombic nature also discovered by DFT, but is at the same time capable of accessing much larger supercells. Based on these calculations, we predict a dielectric constant for LiC_108_ in the range of 9.1–11.9 and recognize the potential of GPrep-DFTB to, for the first time, calculate the dielectric constant of the Li-GIC as a function of the SOC in the near future. Finally, the GPrep-DFTB relaxation of large structures in both dilute and domain-like configurations predicts less expansion of the interlayer distance for the domain structure—again agreeing with previous studies and illustrating the potential of our method for further investigation into the complex and large-scale physics taking place in Li-GICs and for being a new kind of bridge between the atomistic scale and the macroscopic scale of future battery materials.

## Figures and Tables

**Figure 1 materials-14-06633-f001:**
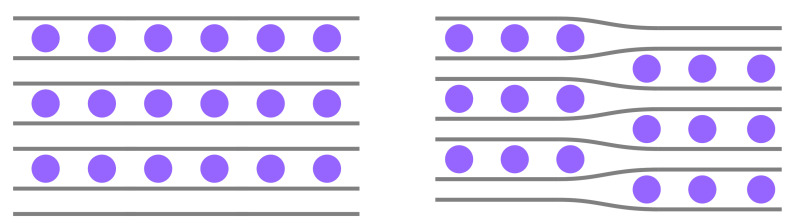
Stage 2 Li-GIC compound (purple: Li-ions, gray: graphene sheets) in a global staging model (**left**) and the Daumas–Heróld domain model (**right**).

**Figure 2 materials-14-06633-f002:**
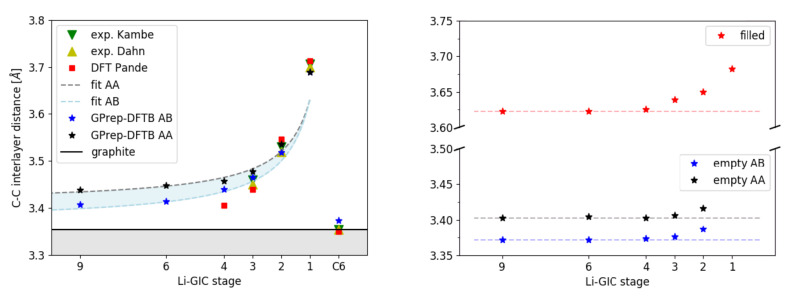
(**Left**): Average C–C interlayer distance in the *z* direction as a function of the stage of the compound in Å. AA and AB signifies AA-stacking and AB-stacking assumed for the empty graphite layers. Where available, experimental and theoretical references are also shown. The curves ‘fit AA’ and ‘fit AB’ correspond to a simple building block model with just two fixed interlayer distances for empty and filled galleries, respectively. For stages 3 and higher, our predictions adhere closely to that model. For stages 1 and 2, the interlayer spacing of the filled galleries was expanded due to the additional charge transfer. (**Right**): Interlayer spacing of the full galleries, as well as empty galleries in AA and AB stacking, as a function of the stage (calculated with GPrep-DFTB).

**Figure 3 materials-14-06633-f003:**
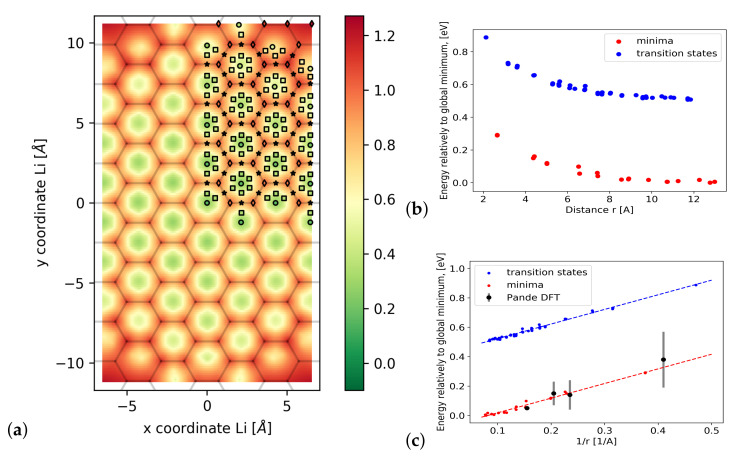
(**a**) Calculated potential energy surface for a freely moving Li-ion with respect to a second fixed ion in the corner of the unit cell (see main text for details of the structural model). Color gradient in eV relative to the global minimum at (0/0). Circles are local minima, diamonds are local maxima, stars are transition states (calculated with the cNEB method), and squares are additional images from the NEBs—all of which have been used for the 2D fit; (**b**) Potential energy depending on the distance between the two Li-ions (blue dots are the transition positions, red dots are the ground states); (**c**) Potential energy depending on the reciprocal distance—clearly showing the Coulombic nature of the interaction, which agrees with [13].

**Figure 4 materials-14-06633-f004:**
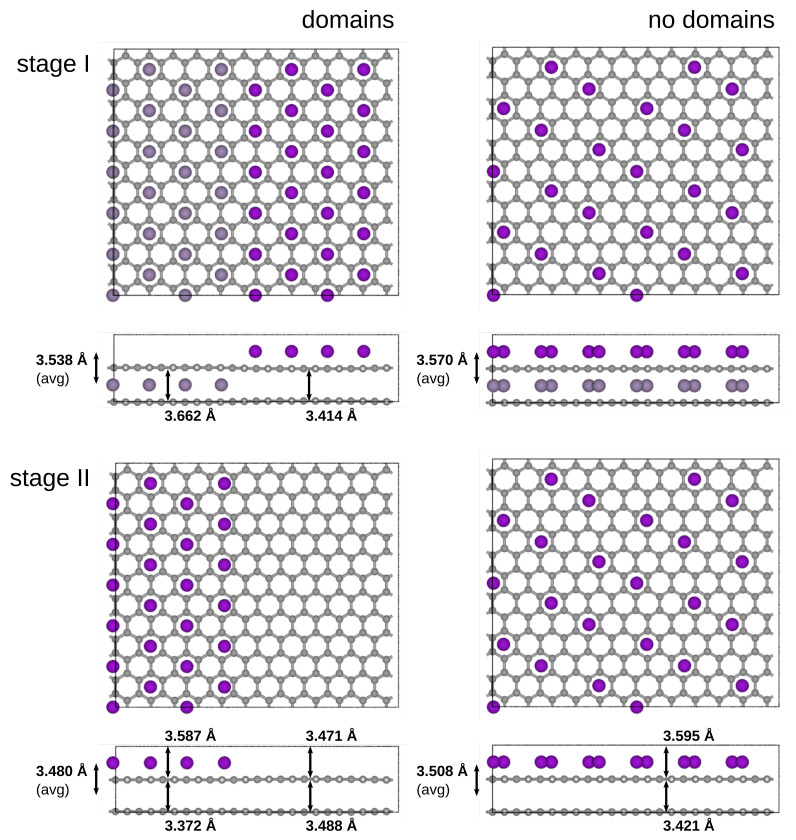
Top- and sideview of stage 1 and stage 2 domain-like and dilute Li-GIC configurations. Structures are fully relaxed, providing both overall average interlayer distances and local interlayer distances for filled and empty areas. For both stages, the domain-like structure has a smaller overall interlayer distance.

**Table 2 materials-14-06633-t002:** C–C interlayer distances in Å. For details, see the caption of Figure 2.

	DFTB			Experiment		DFT
	**Filled**	**Empty AA/AB**	**Avg. AA/AB**	**Avg. Dahn [5]**	**Avg. Kambe [9]**	**Avg. Pande [13]**
stage 1	3.682	-	3.682/3.682	3.700	3.706	3.713
stage 2	3.652	3.416/3.387	3.535/3.518	3.520	3.530	3.546
stage 3	3.639	3.406/3.376	3.478/3.465	3.450	3.460	3.439
stage 4	3.625	3.403/3.373	3.457/3.440	-	-	3.406
stage 6	3.622	3.405/3.372	3.440/3.414	-	-	-
stage 9	3.622	3.403/3.372	3.427/3.407	-	-	-
graphite	-	3.402/3.373	3.402/3.373	3.355	3.355	3.35

**Table 3 materials-14-06633-t003:** Interlayer distances and migration barriers (to the neighboring Li-position), calculated with GPrep-DFTB for a variety of differently stacked, staged, and filled Li-GICs. Regarding the stacking description, *A* refers to the carbon sheets, whereas α and β refer to the ordering of Li-ions.

	Stage	Stacking	In-Plane %	Avg LS [Å]	Filled/Empty [Å]	Barrier [eV]
LiC_48_	III	AAAα	1/3	3.446	3.525/3.406	0.493
Li_2_C_48_	III	AAAα	2/3	3.478	3.616/3.409	0.441
Li_3_C_48_	III	AAAα	3/3	3.478	3.631/3.402	0.424
LiC_36_	II	AAα	1/3	3.469	3.530/3.408	0.504
Li_2_C_36_	II	AAα	2/3	3.512	3.614/3.410	0.451
Li_3_C_36_	II	AAα	3/3	3.535	3.652/3.418	0.426
Li_2_C_36_	I	AαAα	1/3	3.539	3.539/−	0.492
Li_4_C_36_	I	AαAα	2/3	3.625	3.625/−	0.443
Li_4_C_36_*	I	AαAβ	2/3	3.658	3.658/−	0.412
Li_6_C_36_	I	AαAα	3/3	3.682	3.682/−	0.404
Li_6_C_36_*	I	AαAβ	3/3	3.758	3.758/−	0.396

**Table 4 materials-14-06633-t004:** Formation energies in eV per formula unit, calculated with GPrep-DFTB and compared to an overview of the experimental and theoretical results from other studies. Both DFTB and DFT values are variations in the total energy $\Delta E_{\mathrm{int}}$, while experimental values are variations in enthalpy $\Delta H_{\mathrm{int}}$ where available—otherwise they are variations in free energy $\Delta G_{\mathrm{int}}$).

	GPrep-DFTB	Experiment			DFT			
**Ref:**		**(** * **a** * **)**	**(** * **b** * **)**	**(** * **c** * **)**	**(** * **d** * **)**	**(** * **e** * **)**	**(** * **f** * **)**	**(** * **g** * **)**
LiC_6_	−0.14	−0.156 *	−0.114	−0.144	−0.17	−0.22	−0.07	−0.23
LiC_12_	−0.40	−0.228 *	−0.352	−0.257	−0.27	−0.28	−0.12	−0.33
LiC_18_	−0.42	−0.273 *	−0.492	−	−	−	−	−0.29

Experimental references: (a), [19] (b) [6] OCV, vs. solid Li; (c) calorimetry, vs. liquid Li [20]; DFT references: (d) revPBE-D3-BJ [14]; (e) optB88-vdW [12]; (f) revPBE-vdW [12]; (g) GGA-PP [11]; * These values are intercalation free energies ∆Gint. The value for LiC18 is not directly given in the paper but can be consistently extracted using the same formula the authors used for LiC12.

## Data Availability

Data available at https://doi.org/10.5281/zenodo.5636279 (accessed on 27 October 2021).

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
