# Peer review of "Accessing Structural, Electronic, Transport and Mesoscale Properties of Li-GICs via a Complete DFTB Model with Machine-Learned Repulsion Potential"

_materials, 2021, doi:10.3390/ma14216633_

Round 1
Reviewer 1 Report
In this manuscript by Simon Anniés et al., a method based on density functional tight binding (DFTB) fitted to dispersion-corrected DFT data using Gaussian process regression (GPR) is proposed. The long range Li-Li interactions, dielectric constants, domain-formation parameters and others are calculated in order to connect DFT and meso-scale methods for the Li-GIC system. The method correctly quantitively predicts the non-linear nature of the increase of the interlayer distance in LiCx upon intercalation.
The lithium-graphite intercalation compounds (Li-GICs) are actively investigated and improved as the most commercially common anode material. The topic of the study is highly interesting to the readers and the community, in particular, because machine learning for potentials is the state-of-the-art advanced approach. The mechanisms and states of charge are described in detail. The manuscript is well written and organized. My only recommendation is the following. The introduction section might be modified to include more references which appear in the later sections, e.g., [45], [46]. Figure 1 in Supporting Information needs better quality.
Thus, I conclude that this work is a valuable advance in this field, the manuscript is entirely suitable fo publishing in Materials.
Author Response
Reviewer Point 1.0:
In this manuscript by Simon Anniés et al., a method based on density functional tight binding (DFTB) fitted to dispersion-corrected DFT data using Gaussian process regression (GPR) is proposed. The long range Li-Li interactions, dielectric constants, domain-formation parameters and others are calculated in order to connect DFT and meso-scale methods for the Li-GIC system. The method correctly quantitively predicts the non-linear nature of the increase of the interlayer distance in LiCx upon intercalation.
The lithium-graphite intercalation compounds (Li-GICs) are actively investigated and improved as the most commercially common anode material. The topic of the study is highly interesting to the readers and the community, in particular, because machine learning for potentials is the state-of-the-art advanced approach. The mechanisms and states of charge are described in detail. The manuscript is well written and organized.
Reply:
We thank Reviewer 1 for their positive assessment.
Reviewer Point 1.1:
My only recommendation is the following. The introduction section might be modified to include more references which appear in the later sections, e.g., [45], [46].
Reply:
We added the suggested reference as well as some more in the introduction.
Reviewer Point 1.2:
Figure 1 in Supporting Information needs better quality.
Reply:
Indeed. We improved the figure.
Reviewer Point 1.3:
Thus, I conclude that this work is a valuable advance in this field, the manuscript is entirely suitable fo publishing in Materials.
Reply:
Once again we thank Reviewer 1 for their positive assessment.
Note:
All the modifications, including small typesetting changes not listed here, are highlighted in the revised files.
Reviewer 2 Report
The article ‘Accessing Structural, Electronic, Transport and Mesoscale Properties of Li-GICs via a complete DFTB Model with Machine-Learned Repulsion Potential’ reports the use of GPrep-DFTB tool to theoretically study the structure, energetic and electronic properties of Li-GICs in lithium-ion batteries. The study combines theoretical modeling with experimental validation, and benchmarks with reference results to get meaningful conclusions. Moreover, the understanding of the Li-GIC system is very crucial for further improvements of lithium-ion batteries. Therefore, this article is recommended for publication in Materials after addressing the following questions.
- The authors are encouraged to provide the experimental lithiation curves for the graphite and HOPG electrodes and mark the LiCx compounds on the curves.
- The authors stated in line 255 ’Consequently, the values are not intended to be quantitatively comparable, but one may still identify qualitative trends’. What is the benefit of using GPrep-DFTB to identify trends, compared to using DFT?
- In line 268, ‘However, in the regions corresponding to the phase transitions between LiC18 to LiC12 and LiC12 to LiC6, the measured HOPG curve matches within 0.1 V with the references.’ 0.1 V can make the lithiation level significantly different (x value in LiCx) for graphite anode. Is it possible to decrease the difference by tuning the parameters of the model?
Author Response
Reviewer Point 2.0:
The article ‘Accessing Structural, Electronic, Transport and Mesoscale Properties of Li-GICs via a complete DFTB Model with Machine-Learned Repulsion Potential’ reports the use of GPrep-DFTB tool to theoretically study the structure, energetic and electronic properties of Li-GICs in lithium-ion batteries. The study combines theoretical modeling with experimental validation, and benchmarks with reference results to get meaningful conclusions. Moreover, the understanding of the Li-GIC system is very crucial for further improvements of lithium-ion batteries. Therefore, this article is recommended for publication in Materials after addressing the following questions.
Reply:
We thank Reviewer 2 for their positive assessment.
Reviewer Point 2.1:
- The authors are encouraged to provide the experimental lithiation curves for the graphite and HOPG electrodes and mark the LiCx compounds on the curves.
Reply:
We did not initially include the curve because the transitions are not as well resolved as those obtained from e.g. coin cells. This is due to the small insertion surface of HOPG which requires extremely long intercalation times to observe well-defined structural transitions. Work is currently underway to improve the voltage curve. However, the transitions are visible enough to conclude that the HOPG curve substantially agrees with the published curves with different forms of carbon. As such, we included it in the Supplementary Information, for both HOPG and a coin cell.
Reviewer Point 2.2:
- The authors stated in line 255 ’Consequently, the values are not intended to be quantitatively comparable, but one may still identify qualitative trends’. What is the benefit of using GPrep-DFTB to identify trends, compared to using DFT?
Reply:
Both DFTB and DFT are not capable to provide direct quantitative agreement within the simple model of calculating formation energies of ideal stage compounds. However, DFTB is much cheaper than DFT. This on the one hand means that the same qualitative trends are accessible at a fraction of the cost. On the other hand, the bigger advantage is that, thanks to the reduced cost, DFTB allows more extensive calculations (larger supercell, statistical sampling etc.) which will indeed make quantitative predictions possible. As such, we modified the relevant paragraph as follows:
"Consequently, the values given within the scope of this parametrization study are not yet intended to be quantitatively comparable to experiment, but one may still identify qualitative trends, just like with regular DFT. The DFTB model opens up the way to forthcoming more realistic, quantitative simulations."
Reviewer Point 2.3:
- In line 268, ‘However, in the regions corresponding to the phase transitions between LiC18 to LiC12 and LiC12 to LiC6, the measured HOPG curve matches within 0.1 V with the references.’ 0.1 V can make the lithiation level significantly different (x value in LiCx) for graphite anode. Is it possible to decrease the difference by tuning the parameters of the model?
Reply:
0.1 eV was actually a typo. The agreement is within 0.01-0.02 V. We corrected the value.
Note:
All the modifications, including small typesetting changes not listed here, are highlighted in the revised files.